# Risk-Based Mapping Tools for Surveillance and Control of the Invasive Mosquito *Aedes albopictus* in Switzerland

**DOI:** 10.3390/ijerph19063220

**Published:** 2022-03-09

**Authors:** Damiana Ravasi, Francesca Mangili, David Huber, Laura Azzimonti, Lukas Engeler, Nicola Vermes, Giacomo Del Rio, Valeria Guidi, Mauro Tonolla, Eleonora Flacio

**Affiliations:** 1Department for Environment Constructions and Design, Institute of Microbiology (IM), University of Applied Sciences and Arts of Southern Switzerland (SUPSI), 6850 Mendrisio, Switzerland; lukas.engeler@supsi.ch (L.E.); valeria.guidi@supsi.ch (V.G.); mauro.tonolla@supsi.ch (M.T.); eleonora.flacio@supsi.ch (E.F.); 2Department of Innovative Technologies, Dalle Molle Institute for Artificial Intelligence Studies (IDSIA), University of Applied Sciences and Arts of Southern Switzerland (SUPSI), 6962 Lugano-Viganello, Switzerland; francesca.mangili@supsi.ch (F.M.); david.huber@supsi.ch (D.H.); laura.azzimonti@supsi.ch (L.A.); nicola.vermes@gmail.com (N.V.); giacomo.delrio@supsi.ch (G.D.R.)

**Keywords:** *Aedes albopictus*, ovitrap, regularized logistic regression, ecological niche model, environmental factors, surveillance

## Abstract

Background: In Switzerland, *Aedes albopictus* is well established in Ticino, south of the Alps, where surveillance and control are implemented. The mosquito has also been observed in Swiss cities north of the Alps. Decision-making tools are urgently needed by the local authorities in order to optimize surveillance and control. Methods: A regularized logistic regression was used to link the long-term dataset of *Ae. albopictus* occurrence in Ticino with socioenvironmental predictors. The probability of establishment of *Ae. albopictus* was extrapolated to Switzerland and more finely to the cities of Basel and Zurich. Results: The model performed well, with an AUC of 0.86. Ten socio-environmental predictors were selected as informative, including the road-based distance in minutes of travel by car from the nearest cell established in the previous year. The risk maps showed high suitability for *Ae. albopictus* establishment in the Central Plateau, the area of Basel, and the lower Rhone Valley in the Canton of Valais. Conclusions: The areas identified as suitable for *Ae. albopictus* establishment are consistent with the actual current findings of tiger mosquito. Our approach provides a useful tool to prompt authorities’ intervention in the areas where there is higher risk of introduction and establishment of *Ae. albopictus*.

## 1. Introduction

In the last four decades, the Asian tiger mosquito, *Aedes albopictus* (Skuse, 1894), has been spreading globally outside its native range in Southeast Asia, to the point of being listed as one of the world’s worst invasive alien species [1,2]. Its invasion success has been favored by both extrinsic factors, such as increase of global trade and travel, climate change and lack of efficient control, and intrinsic factors, such as strong physiological and ecological plasticity [1,3,4]. For instance, the ability to produce cold-tolerant eggs through a photoperiodic diapause response allows the mosquito to overwinter at the egg stage in temperate climates [5,6].

Thanks to its adaptation to use artificial containers as breeding sites, *Ae. albopictus* thrives in suburban and urban environments [1]. Host-seeking adult females can cause severe nuisance due to their aggressive daytime outdoor biting activity. In addition, their ability to transmit several arboviruses, including dengue, chikungunya, Zika, and yellow fever viruses, raises high public concern [7,8]. The increase of *Aedes*-borne diseases in Europe is becoming quite alarming, with a succession of epidemics of dengue and chikungunya in different countries since the years 2000 [9]. Rising temperatures due to climate change may exacerbate the risk of epidemics by allowing increased mosquito survival, reproduction and biting rate, faster viral amplification, and longer transmission season [10].

In Europe, *Ae. albopictus* appeared for the first time in Italy in the 1990s and then expanded to most areas of the country and to other countries passively through the various human transportation networks (Mosquito Maps, http://ecdc.europa.eu/ (accessed on 16 July 2021)). In Switzerland, its presence was recorded for the first time in 2003 in the southern side of the Alps, in the Canton of Ticino (hereafter referred to as Ticino), at a service area on the European route E35, close to the Italian border [11,12]. Despite containment measures being implemented since its first appearance, *Ae. albopictus* gradually formed permanent overwintering populations and expanded northwards across the lower valleys [13,14]. Today it is considered well established (with reproduction and overwintering) in most urban areas of Ticino (Figure 1).

The spread of *Ae. albopictus* in Ticino has been actively monitored since 2003 by the cantonal Working Group for Mosquitoes (Gruppo Lavoro Zanzare, GLZ) through an extensive surveillance system based on the deployment of oviposition traps (ovitraps) for collection and counting of *Aedes* eggs. Ovitrap data indicate suitability of a habitat for oviposition and therefore for the possibility of establishment of populations. The surveillance has gradually expanded over the years to cover larger areas in response to reported and suspected presence of the mosquito [13]. During the first phases of *Ae. albopictus* invasion in 2003, the surveillance focused on suspected entry points near the border with Italy, such as motorway service areas, high traffic areas, large parking lots, import shops, and freight sorting facilities. Between 2004 and 2008, as the mosquito started to spread and become established at several sites, the surveillance was widened to include industrial areas, larger car parks, and public areas in affected municipalities. From 2009, as the mosquito started to colonize urban areas, the monitoring was extended to an area-wide surveillance network covering the urban areas of entire municipalities, with the adoption of a grid system of 250 m × 250 m units to standardize the distribution of ovitraps on the territory [13]. Currently, more than 80 municipalities are involved in the surveillance and control of the mosquito, covering more than 90% of the total human population of Ticino.

The introduction and spread of *Ae. albopictus* has also been monitored since 2013 at the Swiss national level, along the main potential dispersal routes (i.e., highways, airports, river ports, and freight stations) [15]. To date, *Ae. albopictus* has been observed in the Canton of Grisons, south of the Alps, and is considered established in the southern part of the canton. North of the Alps, the mosquito has become firmly established since 2018 in an area of Basel adjacent to the motorway toll on the Swiss–French border [16]. Small populations of *Ae. albopictus* were also recorded recently in two areas of Zurich: an international bus station located in the center of the city, near the main train station, and a suburban neighborhood in the Wollishofen district (Swiss Mosquito Network, http://www.mosquitoes-switzerland.ch (accessed on 12 October 2021)). Possible foci of introduction have also been observed in the central (e.g., Cantons of Uri and Luzern), northern (e.g., Cantons of Solothurn, Aargau, Basel-Landt, and Basel-Stadt), northeastern (e.g., Cantons of Zurich and Schaffhausen), and western (e.g., Rhone Valley in the Canton of Valais and Cantons of Fribourg, Vaud, and Geneva) parts of Switzerland (Swiss Mosquito Network). These observations, along with the concurrent rapid spread in other European countries, suggest that the tiger mosquito will colonize more and more areas in Switzerland in the near future, introducing the risk of local arbovirus transmission following imported human cases.

Since extensive surveillance might not be economically viable in many municipalities, the early identification of the most suitable areas for spread and establishment of *Ae. albopictus* becomes a critical tool to support the authorities in vector management planning. Ecological niche models can be used to evaluate the potential risk of spread and establishment of invasive species [17]. In particular, correlative distribution models identify relationships between species’ presence-only or combined presence–absence data and socioenvironmental variables (e.g., meteorological data, habitat type classes, and host population density estimates) and use these relationships to make predictions on the potential distribution of the species in unsampled areas of the study region [18]. Temperatures and precipitations have a strong impact on the development and survival of the different *Ae. albopictus* stages [19,20]. Therefore, many models have focused on meteorological variables to predict areas suitable for colonization [4,21]. In Switzerland, remotely sensed land surface temperature data were used by Neteler and colleagues [22] to identify suitable areas for adult survival and overwintering of diapausing eggs and to project climate change induced range shifts of *Ae. albopictus*.

In addition to weather conditions, the distribution of *Ae. albopictus* can also be influenced by other aspects, such as environmental features (e.g., elevation, type of habitat, and vegetation canopy) and anthropogenic factors (e.g., human population density, transport network, and travel), reflecting the impact of human movements and urbanization on the spread of the mosquito [23,24]. In this context, modeling approaches such as machine learning can be helpful to handle complex and multiple interacting elements. Many algorithms have been used to predict the distribution of mosquito species at different spatial scales, from regional [25,26], to national [27,28], continental [3,4,23], and global [29,30], and different variables have been selected by different approaches as the most informative predictors.

Here we used the long-term dataset of *Ae. albopictus* presence–absence records in Ticino to train and validate a regularized logistic regression and to forecast the probability of establishment of *Ae. albopictus*. The most informative explanatory features for the prediction of establishment of *Ae. albopictus* in Ticino were identified. The model was then used to extrapolate the probability of establishment in the whole of Switzerland, for which there are almost no establishment records, and in the cities of Basel and Zurich, where only a small number of occurrences are available for analysis. We created risk maps of *Ae. albopictus* establishment intended to help cantonal and municipal authorities in optimizing efforts and resources by targeting vector surveillance and control in the areas where *Ae. albopictus* has not yet been recorded but where the habitat appears to be suitable for establishment during summer months if mosquitoes are introduced.

## 2. Materials and Methods

### 2.1. Study Area

The study area comprises the entire 41,285 km^2^ territory of Switzerland (45° to 48° N and 5° to 11° E). Switzerland lies in the heart of Europe and is mostly covered by agricultural land (40%), forest and woodland (30%), settlement areas (7.5%), and bodies of water (4%). As a result of its geographical position and its complex topography, the country has a varied climate. The Alps form an important climatic barrier between the north and south of Switzerland. In the Central Plateau, which lies north of the Alps, between the Swiss Jura and the Swiss Alps, the climate is moderately continental, with cold winters often reaching freezing temperatures in January, and warm summers. In the south of the Alps, namely in Ticino, the climate is strongly affected by the Mediterranean Sea, with mild winters and warm and humid summers, sometimes hot. Switzerland is particularly affected by climate change. Since measurements began in 1864, the average temperature has already increased by approximately 2 °C [31]. This is about twice the average global temperature rise. The five warmest years in the measurement series 1864–2019 were all recorded after 2010. Other evidence of climate change are heatwaves, more hot days and nights, and shrinking snow cover on the Central Plateau. The frost days have decreased by up to 60% since 1961. Heavy precipitation events have also become more intense and more frequent [31].

The overall area of Switzerland was rasterized into 1,465,516 square pixels (or cells) with a 200 m resolution based on the freely available digital height model of Switzerland DHM25/200 m with coordinate system CH1903/LV03 (EPSG: 21781) LN02 (EPSG: 5728), produced by the Swiss Federal Office of Topography [32]. The digital height model DHM25/200 m is a dataset representing the three-dimensional form of the Earth’s surface without vegetation or built-up areas. All data layers used in the model were matched to this grid using QGIS 2.18.0 and ArcGIS 10.2 (ESRI, Redlands, CA, USA).

### 2.2. Response Variable

Data for *Ae. albopictus* in Ticino are available through the long-term surveillance system based on the deployment of ovitraps [13,33]. The number of ovitraps has increased during the years, from 57 traps in 2004 to a maximum of 1389 traps deployed in 2013. The ovitraps are deployed at the beginning of the active season of the mosquito, in May, and are inspected every two weeks until the end of September. At each inspection round, the wooden slat of each ovitrap, where mosquitoes lay their eggs, is collected and replaced with a new one. In the laboratory, the slats are analyzed under a stereo microscope (EZ4 D, Leica Microsystems, Wetzlar, Germany), and eggs of container-breeding *Aedes* are examined as described in previous work [14]. Before 2015, only two container-breeding *Aedes* species were present in Ticino: the indigenous species *Ae. geniculatus* and the invasive species *Ae. albopictus* [13,14]. The eggs of these two species can be morphologically differentiated under a stereo microscope, and eggs of *Ae. albopictus* were counted. As of 2015, two new invasive mosquito species, namely *Ae. koreicus* and *Ae. japonicus,* are present in different areas of Ticino [15]. Since the eggs of these two species cannot be morphologically differentiated from the eggs of *Ae. albopictus* without resorting to special microscopy equipment and expertise, data collected after 2015 were not considered for this study.

Data from each ovitrap, including geographic coordinates, date of collection, and number of *Aedes* eggs, is stored in a national database managed by the Institute of microbiology and the info fauna—CSCF (http://www.cscf.ch/ (accessed on 10 January 2022)).

Since the wooden slats are deployed in the field for periods of two weeks, the yearly surveillance period was divided in ten periods of two weeks (from period 1 including calendar weeks 20–21 to period 10 including calendar weeks 38–39). In order to assess the stable presence of *Ae. albopictus* in an ovitrap each year, we defined an establishment indicator. We say that establishment for an ovitrap occurred when a sequence of consecutive positive wooden slats was observed with the last positive slat occurring more than 28 days after the first one. The period of 28 days was defined in order to account for the fact that from 2013, slats were analyzed at alternate rounds (i.e., every 28 days) [14]. Subsequently, each ovitrap was assigned, based on its geographical position, to the appropriate 200 m × 200 m cell of the model grid. *Aedes albopictus* was considered established in a cell in a given year if there was establishment in at least one of the ovitraps located in the cell. Thus, the response variable Established was defined as 1 (TRUE) if there had been establishment in a cell in a given year, and 0 otherwise. This resulted in a total number of 7268 establishment/non-establishment data (Table 1). Figure 1 shows the occurrence of *Ae. albopictus* from 2005 to 2015.

Establishment data for years 2005 to 2012 were used to train the model. Data from 2004 were not used in the model but were used, as described later, to calculate one of the predictors (namely car distance to establishment), which requires establishment data from the previous year to be computed. The training dataset of 4282 observed cells was composed of 990 established cells and 3292 non-established cells. The model validation was performed with 2961 cells for years 2013 to 2015 comprising 1608 established cells and 1353 non-established cells (Table 1).

### 2.3. Predictors Used for Ae. albopictus Habitat Suitability

A total of 79 candidate predictors were selected based on their potential relevance to *Ae. albopictus* distribution (Appendix A). These included static attributes for cell terrain morphology (*n* = 3), type of land cover (*n* = 32), and permanent host population (*n* = 1), and dynamic features such as meteorological variables (*n* = 42) and travel distance from the nearest cell with establishment (*n* = 1).

Three terrain morphology features were derived from the digital height model of Switzerland DHM25/200 m: cell elevation, aspect (i.e., land orientation or the compass direction that a topographic slope faces), and slope. These topographic features give indirect measures of the climate at the cell’s scale and can influence the biotic conditions of mosquito and the choice of suitable breeding sites.

The way a habitat type is distributed relates directly to habitat preferences in mosquitoes. Land cover data were obtained from CORINE Land Cover 2012 (CLC-CH 2012), where images acquired by Earth observation satellites in 2011–2012 were used as the main source data to derive land cover classes (© European Union, Copernicus Land Monitoring Service 2012, European Environment Agency; [34]). In Switzerland, CORINE identifies 32 types of land cover, such as artificial surfaces, agricultural areas, forests, semi-natural areas, wetlands, and water bodies (Appendix A). Percentages of coverage for each class were extracted for each 200 m × 200 m cell.

The total human host population was extrapolated from the layer STATPOP2016 (Swiss Federal Statistical Office, GEOSTAT), based on 2016 statistical data of the total permanent human resident population. The punctual data of the STATPOP2016 100 m × 100 m grid were forced on the reference 200 m × 200 m grid and the total population in each cell was calculated.

Meteorological predictors were derived from the MeteoSwiss spatial climate daily datasets (source: MeteoSwiss, Zurich-Airport, Switzerland). The temperature datasets are constructed through interpolation of daily minimum, maximum, and mean temperatures from a weather station network of approximately 90 weather stations (measuring free-air temperature 2 m above ground level) to a 1 km resolution grid in the Swiss coordinate system CH1903 [35,36]. The precipitation dataset is built from daily precipitation totals measured at the MeteoSwiss high-resolution rain-gauge network [37].

The daily datasets were processed with Apache Spark^TM^ 2.44 (The Apache Software Foundation, Wilmington, DE, USA) [38] to compute seasonal predictors. Cold season (cold-s) for a given year *y* included the months from the beginning of December of the previous year (*y*-1) to the end of February. These months represent the period in which diapausing mosquito eggs in Switzerland are subjected to the coldest temperatures in winter [14]. Warm season (warm-s) lasted from the beginning of May to the end of September. These months represent the period of the year in which *Ae. albopictus* is reproducing in Switzerland [14]. We calculated the seasonal average (avg) and the percentiles 5, 25, 75, and 95 (named p5, p25, p75, and p95, respectively) for the following data series: daily minimum, maximum, and mean temperatures (named Tmin, Tmax, Tmean, respectively) and daily rainfall (named RAIN). Percentiles 5 and 25 were not used for RAIN (both warm and cold seasons) as they are constantly equal to zero. In total, we obtained 36 meteorological seasonal predictors.

To account for meteorological conditions on a more refined temporal scale, we also included some features based on two-week statistics of MeteoSwiss data series. These features account better for prolonged extreme meteorological conditions than seasonal predictors. Namely, Tmin minimum biweekly maximum and Tmin minimum biweekly average are defined as the minima of the maximum and average values of Tmin over any two-week period of the year and are introduced to measure how cold two of the coldest two-week periods of each year were; features Tmax maximum biweekly minimum and Tmax maximum biweekly average are defined as the maxima of the minimum and average values of Tmax over any two-week period of the year and are introduced to measure how hot two of the hottest two-week periods of each year were; warm-s RAIN maximum biweekly average and warm-s RAIN minimum biweekly average are defined as the minimum and the maximum of the average values of RAIN over any two-week period of the warm season and are introduced to measure the amount of rainfall in one of the, respectively, wettest and driest periods of the warm season. The reader is referred to Appendix B for a more detailed description of these features.

Since the coordinate system of the MeteoSwiss grid matched the one of the digital height model DHM25/200 m, except for the lower resolution, the meteorological predictors’ layer was overlapped to the DHM25/200 m.

In order to account for the propagation paths of adult mosquitoes in neighborhood areas, we also included in the set of predictors the distance of each cell from the closest cell in which mosquitoes had established the year before. As mosquitoes can be displaced by means of passive transport (especially cars) in their adult forms [39], distance was measured by the travel time in minutes necessary to go from the center of one cell to the center of the other. For each cell *c* at year *y*, the travel time from all cells established at year *y*-1 to cell *c* was computed and the minimum value obtained was retained as the value of the feature car distance to establishment, thus estimating the minimum duration of a passive transport of mosquitoes to cell *c*. The duration of car travels was estimated using the Open Source Routing Machine (OSRM, http://project-osrm.org/ (accessed on 16 March 2021) [40]). OSRM is a free C++ implementation of a high-performance routing engine for shortest paths in road networks combining routing algorithms with the open and free road network data of the OpenStreetMap project (OSM, https://www.openstreetmap.org (accessed on 16 March 2021) [41]). Notice that the car distance to establishment feature for the years 2005 to 2015 is based on the values of feature Established (response variable) of years 2004 to 2014.

### 2.4. Predictive Model

Established and non-established data were modeled using an ensemble of regularized logistic regression models to predict *Ae. albopictus* establishment probability. By aggregating multiple different models, the ensemble approach tries to compensate the weaknesses of individual models to produce a more robust prediction [42]. The ensemble included eight models learned from different training datasets using the logistic regression methods included in the scikit-learn Python package 0.24 [43]. Logistic regression assigns a weight (coefficient) to each predictor and returns the probability that variable Established takes value TRUE. Variables were preprocessed by min–max normalization, which consists of rescaling their range to the interval [0, 1]. Most of the predictors considered were correlated and, therefore, redundant. For this reason, we adopted a regression approach with a strong penalization: the least absolute shrinkage and selection operator (LASSO) [44] was used for predictors’ selection and regularization to avoid parameter overfitting. LASSO is a penalized least square method that shrinks the coefficient space by imposing an L1 penalty (sum of absolute values) on the regression coefficients which are finally set to zero for the least informative variables, thus removing them from the model. This means that the predictors’ selection was performed automatically while learning the model, based on a regularization parameter *C* representing the strength of the penalization (higher values of *C* implying a smaller number of selected predictors). Different values of *C* were assessed using a cross validation over the individual models of the ensemble, as described below. In a preliminary analysis, we found that inside the range [0.01, 1] the choice of *C* was not significantly affecting the performance. We therefore chose the intermediate value of 0.1, which produced models selecting between nine and 14 most relevant features; overall, only 17 variables are selected by at least one model.

The eight models of the ensemble were obtained as follows. Ticino data from 2005 to 2012 were divided into eight folds and each model was trained using only seven folds, using a leave-one-year-out cross-validation strategy. Therefore, rather than randomly splitting the data into *k* folds, we grouped them by year to increase the diversity among models. We thus obtained eight models, each one learned from a subset of the training data obtained by leaving out a different year. Finally, to improve the overall predictive performance and provide an estimate of the prediction uncertainty, we used the eight models within an ensemble approach producing a single ensemble prediction and a measure of the prediction uncertainty, respectively, as the mean and the standard deviation of the eight models’ predictions.

The performance of the ensemble model was evaluated using the area under the receiver operating curve (AUC). The AUC represents the probability that, given two random cells, one established and one non-established, the classifier assigns a larger risk of establishment to the established cell. This gives us a measure of how good the classifier is in ordering the suitability of different areas to *Ae. albopictus* establishment. All data from 2007 to 2015 were used for validation. Years 2005 and 2006 were excluded due to lack or very small number of established cells. Since data from 2007 to 2012 were also used for training the models, for this period, model assessment was performed by cross validation: the AUC of each individual model of the ensemble was computed on the data from the year left out from the training set. Models leaving out years 2005 and 2006 were therefore not considered in this first cross-validation assessment. The adopted strategy, besides producing more diverse models, avoids overestimating the performance by excluding data from the same year as the test set from the training set. In fact, the similarity between cells both in terms of meteorological condition and establishment can be expected to be larger within the same year than across different years. In a second assessment, we compared the prediction of the probability of establishment (hereafter also referred to as risk of establishment) of *Ae. albopictus* in Ticino based on the predictor features of the years 2013, 2014, and 2015 to the actual data of establishment for those years, both for each individual model and for the ensemble prediction.

Due to the LASSO penalization adopted, the eight trained models assigned a null coefficient to the features that granted no improvement to the outcome of the prediction and, thus, ended up using only a subset of the initial features. This allowed us to identify the most informative explanatory features for the prediction of the distribution of *Ae. albopictus* in Ticino by looking at the proportion of individual models of the ensemble assigning them a non-zero coefficient.

The ensemble model was then used to create maps estimating the risk of *Ae. albopictus* establishment in Switzerland. For each 200 m × 200 m cell of the whole Switzerland, we computed the predictive features and input them to the model to predict the probability of establishment. Since data about establishment in the past were available only for Ticino, it was not possible to compute the car distance to establishment feature in the rest of the country. Therefore, we kept a fixed car distance to establishment of 0.5 min (30 s) for all cells, implying that establishment has already occurred in the neighborhood. This provided an overview of the areas in Switzerland which are mostly at risk in case a first colony of mosquitoes arrives.

We then focused on two areas, the cities of Basel and Zurich, for which we did have records of positive ovitraps for 2019 (37 points of first establishment for Basel and 16 for Zurich; data kindly provided by the Inspection body for chemical and biosafety (KCB) of the Cantonal Laboratory of Basel-Stadt, the Swiss Tropical and Public Health Institute, the Urban Pest Advisory Service of the City of Zurich, and the Office for Waste, Water, Energy and Air (AWEL) of the Canton of Zurich). In this case, it was possible to build the car distance to establishment feature necessary to build the risk maps by computing the smallest distance between each cell of Basel or Zurich and the identified position of 2019 establishments.

In order to build suitability maps of *Ae. albopictus* establishment, it was also necessary to create scenarios for the meteorological conditions of the coming years. For this, we decided to use meteorological data from the years 2015, 2016, 2017, and 2018 in order to analyze multiple realistic scenarios. We finally produced two types of maps, by projecting probabilities of establishment onto Swiss national maps (Federal Office of Topography swisstopo): a suitability map showing the average of the 32 establishment probabilities predicted by the eight models of the ensemble for the four meteorological scenarios (years 2015 to 2018) and an uncertainty map representing the standard deviation of the 32 predictions, which provides an indication about the reliability of risk estimate displayed in the suitability map.

## 3. Results

### 3.1. Informative Predictors

The importance of the predictors was evaluated based on the coefficients assigned to them by each of the eight LASSO logistic regression models trained in cross validation on years 2005 to 2012. Table 2 shows the predictors that were assigned a non-null coefficient by at least half of the models of the ensemble and the sign of the assigned coefficients. Each feature is ranked based on its frequency, i.e., the proportion of models using it as a predictor of the establishment probability. The more models that select a predictor, the more likely it is that it provides useful information about mosquitoes’ establishment. The sign of the coefficients indicates the direction of the relation. For instance, the negative sign of the coefficient associated to elevation means that the higher the elevation of a 200 m × 200 m cell, the lower the probability of *Ae. albopictus* establishment (Table 2). The predictors selected by the ensemble model, their correlation with other variables, and their contribution to the prediction are analyzed in more detail in Appendix C.

Amongst all 79 candidate predictors, only ten were considered informative by at least half of the eight cross-validation models, and thus most significant for predicting the probability of *Ae. albopictus* establishment in Ticino. Figure 2 shows the values of the coefficients assigned to these features by the eight models. It can be noticed that coefficient values are consistent across models. Six of the ten predictors showed a negative relation to the probability of establishment: elevation, the fifth percentile of the minimum temperature in the warm season (warm-s Tmin p5), the average of precipitations in the warm season (warm-s RAIN average), the road-based distance in minutes of travel by car from the nearest cell established in the previous year (car distance to establishment), the percentage of industrial or commercial units covering a cell (clc121 (industrial) land cover), and average daily minimum temperature (Tmin) observed during the two-week period of the year with the lowest average of Tmin (Tmin minimum biweekly average). On the contrary, the 25th percentile of the maximum temperature in the cold season (cold-s Tmax p25), the average precipitation observed during the two-week period of the warm season with the lowest average precipitations (warm-s RAIN minimum biweekly average), the 75th percentile of the precipitations in the cold season (cold-s RAIN p75), and the 75th percentile of the maximum temperature in the warm season (warm-s Tmax p75) showed a positive relation to the probability of establishment.

### 3.2. Model Outputs

The performance of the ensemble was firstly assessed computing the average and standard deviation of the AUC of six individual models of the ensemble on the left-out year of the training dataset. Models leaving out years 2005 and 2006 were excluded due to lack of, or very small, number of established cells in those two years. Secondly, to better validate the ensemble model, the AUC of each individual model was assessed on the establishment data of years 2013, 2014, and 2015 separately. The cross-validation yearly means and standard deviations of the AUCs obtained by the individual models of the ensemble are given in the first row of Table 3. The second row gives, instead, the AUC of the ensemble prediction, i.e., first the individual model predictions are averaged, then the AUC of the average prediction is computed (not meaningful in the case of cross validation). We observe that the AUC obtained by the cross validation over the years 2007 to 2012 (i.e., 0.856) is better than that obtained on the more recent years used for testing (years 2013 to 2015, AUCs from 0.670 to 0.748) when the establishment was more widespread.

The ensemble model was used to predict the probability of *Ae. albopictus* establishment in the whole of Switzerland and in the cities of Basel and Zurich for each of the four scenarios represented by the meteorological conditions observed in years 2015, 2016, 2017, and 2018. We thus obtained 32 estimates of the probability of establishment generated by the eight individual models of the ensemble over the four climate conditions considered for each 200 m × 200 m cell of the areas of interest. Using such predictions, we generated five risk maps for each area: four yearly risk maps with the average of the eight risk estimates obtained for each of the four years (Appendix A, left column) and a summary risk map, showing the overall average of the 32 risk estimates (Figure 3, Figure 4 and Figure 5, top maps).

Each risk map is accompanied by an uncertainty map showing the normalized standard deviation of the predictions used in the corresponding map (Appendix A, right column and Figure 3, Figure 4 and Figure 5, bottom maps). For consistency, the same min–max normalization is applied to all uncertainty maps: original values are rescaled so that the range [0, 1] of the normalized standard deviations corresponds to the original range [0, 0.38], where 0.38 is the maximum standard deviation obtained across all maps. Uncertainty maps describe the reliability of the prediction, higher values of the standard deviation meaning that the prediction is less reliable. We notice, for instance, a red spot in the northeastern part of Switzerland in the 2015 standard deviation map of Appendix A, meaning that, for the peculiar meteorological conditions of 2015, the risk estimated in that area nearby Lake Constance has a large variability and, therefore, should not be trusted.

## 4. Discussion

We developed a regularized logistic regression that used the historical entomological *Ae. albopictus* data from ovitraps in Ticino from 2005 to 2012 as response variable in the training process. In the validation process, the probability of establishment of *Ae. albopictus* in Ticino was predicted both by cross validation over the training years 2007 to 2012 and for the test years 2013, 2014, and 2015 and the entomological data for the same years was used to test the performance of the ensemble model separately for each year. The best performance was obtained in cross validation on data from 2007 to 2012, with an AUC of 0.856, whereas establishment was more difficult to predict for test years from 2013 to 2015 (AUCs between 0.67 and 0.75). We believe that this is due to the fact that, whereas in the initial phase of *Ae. albopictus* colonization the distance from already established colonies is a good predictor of new establishments, in more recent years where mosquitoes are well settled, establishment in specific areas is less related to the nearby situation and more affected by factors, such as possible treatments against the vector or specific features of the traps and traps’ environment, that were not observed. However, since our interest is to predict the spread of mosquitoes in areas of Switzerland where they are not yet settled, an AUC of 0.856 on the years of mosquitoes’ diffusion allows us to expect the probabilities of establishment predicted by the ensemble in unobserved cells to be quite informative about the most probable directions of *Ae. albopictus* further diffusion.

When building suitability maps of *Ae. albopictus* establishment, we used meteorological data from years 2015, 2016, 2017, and 2018 as scenarios for the meteorological conditions of the coming years. Temperature-wise, these years are similar. They represent some of the warmest years in Switzerland since the first measurements in 1864, all of which have been recorded after 2010 [31]. Their average annual temperature exceeds the average of the years 1981–2010 by 0.7 to 1.5 °C, with 2018 being the hottest year since the beginning of measurements in 1864. These scenarios can also be considered representative of the most recent years 2019 and 2020 and probably also realistic for the coming decade. Indeed, the average annual temperatures of 2019 and 2020 follow the same trend, exceeding the average of the years 1981–2010 by 1.1 and 1.5 °C, respectively. Differently, no change in average summer precipitation has been discerned in Switzerland so far. Nevertheless, there are clear signs that heavy precipitation pattern is slowly changing, with intensity and frequency of heavy rainfall increasing in all seasons and regions of Switzerland.

Of the 79 candidate predictors (including terrain morphology, type of land coverage, total human population, meteorological indicators, and travel distance from the nearest established cell) contemplated as potentially relevant for predicting the probability of *Ae. albopictus* establishment in Ticino, only ten were considered informative by at least half of the eight ensemble models. Interpreting data-driven models can be hard and misleading, especially when features are correlated, as in this case. However, we hereafter analyze the parameters of the ensemble model and try to provide an interpretation for them with the aim of suggesting hypotheses that might stimulate the discussion and inspire further research.

Altitude, which is regarded as a proxy of temperature, contributed significantly to the prediction of establishment: the higher the elevation of a cell is, the lower the probability of *Ae. albopictus* establishment will be. The effect of altitude has been well observed in Ticino, where the establishment of the mosquito is delayed above 400 m.a.s.l. [14]. Out of 30 temperature predictors at a seasonal temporal scale and four temperature predictors at a two-week temporal scale, only four were considered informative. Two of them account for temperatures in the warm season: the fifth percentile of the minimum temperature in the warm season (warm-s Tmin p5) shows a negative association with the probability of establishment, while the 75th percentile of the maximum temperature in the warm season (warm-s Tmax p75) showed a positive relation, the strongest effect being the one of warm-s Tmin p5. The 5% of smallest Tmin can be expected to refer to spring or autumn temperature, whereas the 75% of highest values of Tmax can be expected to happen during the summer. The learned model is therefore saying that somehow colder spring or autumn season as well as hotter summers may favor *Ae. albopictus* establishment. The direction of warm-s Tmin p5 association with the establishment moves in the opposite direction than the one we had expected, since high temperatures are, in general, expected to increase the suitability of an area to *Ae. albopictus* establishment. However, the warm-s Tmin p5 feature might indirectly account for the effect of other non-observed variables, e.g., humidity, with higher values of warm-s Tmin p5 being possibly associated with drier conditions in spring or autumn.

Two other selected predictors account for cold-season temperatures: the 25th percentile of the maximum temperature in the cold season (cold-s Tmax p25), which showed a positive relation to the probability of establishment, and the average minimum temperature of one of the coldest two-week periods of the year (Tmin minimum biweekly average), which showed a negative relation but also contributed much less than cold-s Tmax p25 to the probability of establishment. The model therefore describes a situation in which *Ae. albopictus* is more likely to establish in the case of warmer winters, although, surprisingly, the probability of establishment is slightly reduced when the temperatures observed during protracted cold periods are higher. Perhaps, the lack of consensus in these two latter predictors is not surprising, since the response variable Established is based on the reproductive success in the warm season, more than on the overwintering success of diapausing eggs. Nonetheless, the pattern of invasion of *Ae. albopictus* in Ticino [13,14] suggests a connection between the establishment and overwintering: if the mosquitoes are established in an area, it is very likely that they will start to overwinter in the area and that they will be present the following years as well.

Precipitation can have both a positive effect on the larval carrying capacity of breeding sites and a negative effect on the mosquito flying activity and reproductive cycle, interrupting it by washing away aquatic stages from container breeding sites [45,46]. This ambivalent effect was indeed detected by our model. The importance of the occurrence of rainfall was highlighted by the positive relation between the average rain observed during the least rainy two-week periods of the warm season (warm-s RAIN minimum biweekly average). However, too much rainfall might have a negative impact on the survival of aquatic stages or the abundance and distribution of mosquito breeding habitats, as indicated by the negative relation of the average of precipitations in the warm season (warm-s RAIN average) with the probability of establishment. This could be due to different reasons such as the increase of breeding sites other than ovitraps (and therefore less recovery of eggs in the ovitraps), the decreased activity of females, or the larvae being washed away by intense precipitations. The latter option might increase in importance in the future with the expected increase in heavy rainfall events. The results also suggested some effect of rainfall in the cold season, with a positive impact of the 75th percentile of the precipitations in this part of the year (cold-s RAIN p75).

The presence and abundance of resident humans does not seem to play a role in the probability of establishment, but the feature indirectly related to humans’ movement, car distance to establishment, was selected as one of the key drivers of *Ae. albopictus* spread. This predictor measured the road-based distance in minutes of travel by car from the nearest cell established in the previous year and showed a significant negative relation to the probability of establishment: the higher the distance to nearest established cell is, the lower the probability of establishment will be, and vice versa. This is, to our knowledge, the first time that this variable is evaluated in a model on *Ae. albopictus* dispersal. Yet, the existence of passive dispersal of adult tiger mosquitoes in private vehicles at both local and medium-range scales has been confirmed in other settings [39], and its importance in Switzerland is supported by the findings of positive ovitraps along the major transportation routes [13,15].

The percentage of industrial or commercial units covering a cell showed a negative relation to the probability of establishment of *Ae. albopictus*: the higher the percentage is, the lower the probability of establishment will be. In Ticino, as observed by Flacio et al. [14], industrial areas in several municipalities were apparently colonized by the mosquito before residential areas with a lead time of two to three years. Indeed, industrial areas may function as key ports (transport of goods and working places of commuters from areas where the mosquito has already established) of passive mosquito introductions. However, later on, the mosquito was more present in the residential than in industrial areas, since, perhaps, more breeding sites are available in residential areas.

The validated model was used to extrapolate the probability of establishment on the whole Swiss territory with promising results. The forecasts are similar among the different years analyzed (Appendix A). In 2016, the areas at risk of establishment are the same as in the map of mean prediction 2015–2018 (Figure 3) but the risk is accentuated, indicating that this year seemed to be more favorable to the establishment of *Ae. albopictus*, and the uncertainty of the ensemble model estimates was low. The areas identified by the model as suitable for *Ae. albopictus* establishment are consistent with the actual findings of ovitraps positive for tiger mosquito. The maps show high probability of establishment along the Central Plateau, a geographical region that stretches from Lake Geneva in the southwest to Lake Constance in the northeast (Appendix A). This region is characterized by a flat to hilly landscape (with an average altitude between 400 and 700 m.a.s.l.), compared to the other more mountainous regions of Switzerland, and is also the most densely populated region of Switzerland. Correspondingly, positive ovitraps have been observed, from southwest to northeast of the Central Plateau, in the cantons of Geneva, Vaud, Fribourg, Solothurn, Luzern, Aargau, Zurich, and Schaffhausen (Swiss Mosquito Network and info fauna—CSCF). High risk of establishment has also been identified in the northern part of the canton Basel-Landt, where positive ovitraps have been recorded, and in the canton Basel-Stadt, where an established colony and several foci are present in the city of Basel. At the same time, a high risk of establishment is predicted in the lower Rhone Valley in the Canton of Valais. Interestingly, the high degree of risk predicted in Geneva and in the lower valleys of Valais, although with a certain degree of incertitude among the models, is very similar in color to the degree of risk predicted in Ticino, where the tiger mosquito is already established and where the ensemble model was validated with and AUC of 0.85. Mosquito-wise, the main difference between Geneva/Valais and Ticino is that in Geneva/Valais the introduction pressure from adjacent established areas is much lower at present. Meanwhile, in nearby France, *Ae. albopictus* is expanding northwards [47] and might soon become a source of mosquitoes for these Swiss regions. Thus, our results suggest that surveillance in the high-risk areas should be increased.

We compared our predictions with those of a previous prediction for invasion of *Ae. albopictus* in Switzerland produced in 2013 with a different approach. The previous model did not use entomological data but instead suitable areas for adult survival and overwintering of diapausing eggs were calculated analyzing remotely sensed land surface temperature data recorded by the MODIS satellite sensors based on set climatic thresholds [22]. The predictions of suitability for adults for 2011 and 2035, which were based on a climate change scenario, are very similar to our predictions for establishment, which were obtained using historical tiger mosquito data instead of climatic thresholds.

The model was also used to extrapolate the probability of establishment in the cities of Basel and Zurich, where only a small number of occurrences is available for analysis (Figure 4, Figure 5, Appendix A). For all years analyzed, Basel seems to be at higher risk of establishment than Zurich, although the uncertainty of prediction (standard deviation of the models) is, in general, higher in Basel than in Zurich. In 2016, a year considered favorable for the establishment of *Ae. albopictus* in Switzerland, the probability of establishment for Basel is similar to the other years, while it appears lower for Zurich. Overall, we can conclude that the city of Basel might need to intensify the surveillance for *Ae. albopictus* foci of introduction, perhaps adopting a grid system to standardize the distribution of ovitraps on the territory, as has been carried out in Ticino [13]. The establishment in the city of Zurich seems less probable, although the present model did not consider yet the potentially positive effect of microhabitats such as catch basins on the probability of overwintering of diapausing eggs [48]. Thus, it appears that the surveillance of *Ae. albopictus* in Zurich could still focus on the hot spots of introduction, such as the international bus station in the city center and the Wollishofen neighborhood where tiger mosquito adults were observed. In 2021, after three years of surveillance and control including reduction of breeding sites, larvicide treatment of catch basins, information of the residents and door-to-door visits, ovitraps were no longer positive for *Ae. albopictus* in the Wollishofen neighborhood [49]. There is a high likelihood that *Ae. albopictus* was successfully eradicated in this area, which is a very rewarding and encouraging outcome for authorities and scientists.

## 5. Conclusions

We developed a model that learns from the historical occurrence data of *Ae. albopictus* in Ticino, Switzerland, acquired through years of surveillance, and predicts the probability of establishment of the mosquito on the area. The model can be used to extrapolate the probability of establishment outside the training area (i.e., whole Switzerland), where the mosquito is not yet established, with promising results. We can see from the predictions that various parts of Switzerland are at risk for establishment of *Ae. albopictus*. Additionally, the model can also be used to create scenarios of establishment and spread in smaller specific areas where there are already localized colonies, as in the city of Basel, or small foci of introduction, as in the city of Zurich. The risk maps with the probability of establishment of *Ae. albopictus* can help cantonal and municipal authorities in optimizing efforts and resources by targeting vector surveillance and control in the areas with higher risk of establishment.

## Figures and Tables

**Figure 1 ijerph-19-03220-f001:**
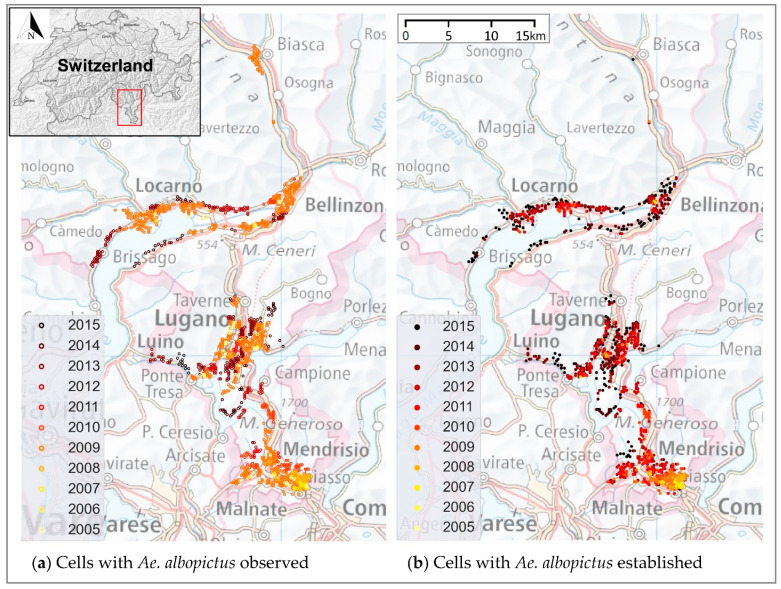
Occurrence of *Ae. albopictus* in Ticino, Switzerland, from 2005 to 2015. (**a**) The empty dots represent observed cells; (**b**) the solid dots represent cells with establishment of the mosquito. Refer to methods for definition of observed and established cells. Maps modified from https://map.geoadmin.ch/ (accessed on 24 February 2022).

**Figure 2 ijerph-19-03220-f002:**
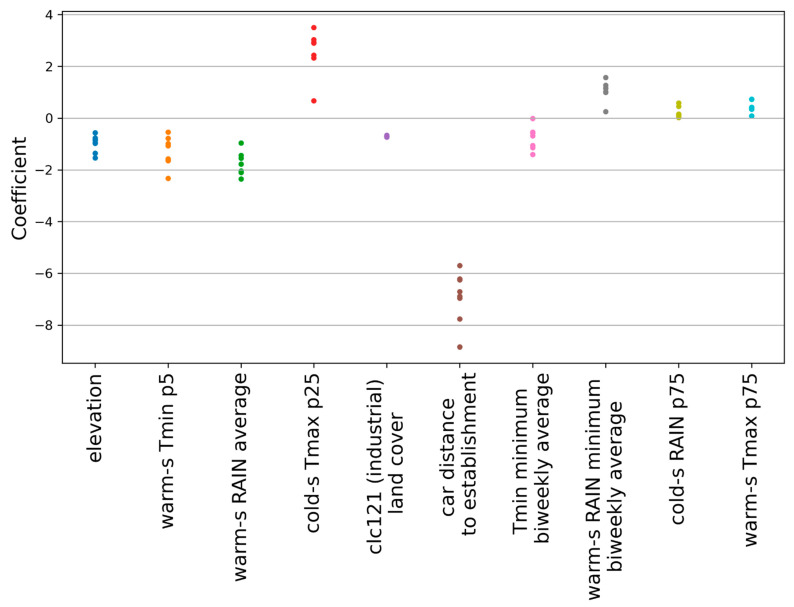
Coefficients assigned by the ensemble models to the mostly used features (i.e., features used by at least half of the models).

**Figure 3 ijerph-19-03220-f003:**
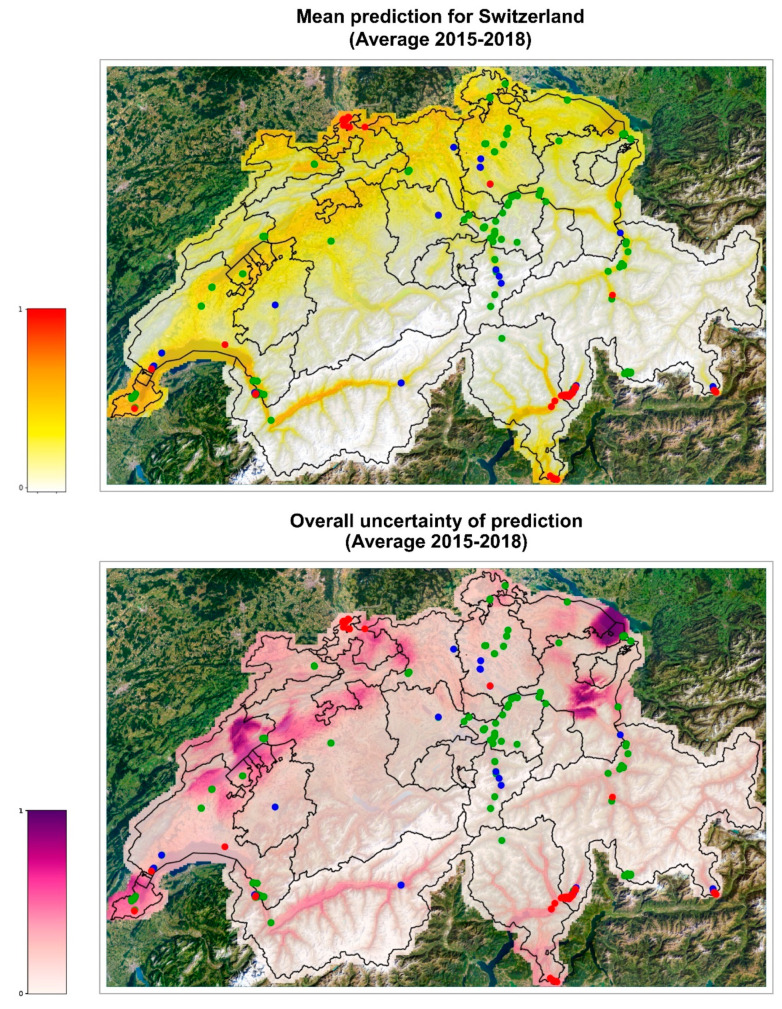
Probabilities of establishment of *Ae. albopictus* in Switzerland. The map on top shows the average risk estimate over the years 2015–2018. The color gradient shows the probability of establishment from 0 (white) to 1 (red). The bottom map represents the uncertainty of the prediction (higher values representing more uncertain predictions). Dots show cells where the presence of *Ae. albopictus* was monitored in 2021, the color green representing cases where it was found absent from the cell, blue where it was present but not established, and red where it was established (data source: Swiss Mosquito Network and info fauna—CSCF). Maps modified from https://map.geoadmin.ch/ (accessed on 24 February 2022).

**Figure 4 ijerph-19-03220-f004:**
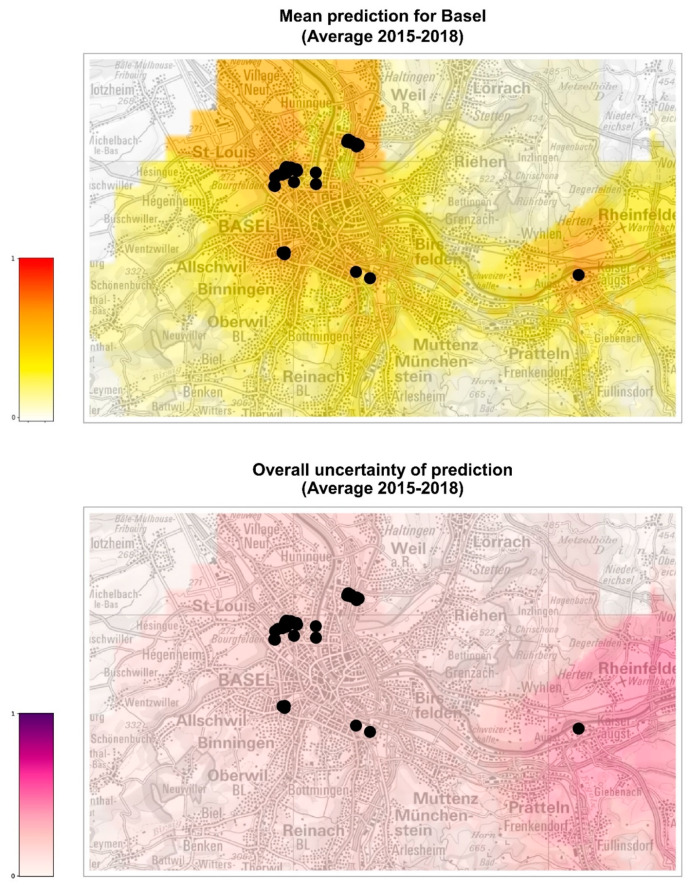
Probabilities of establishment of *Ae. albopictus* in Basel. The map on top shows the average risk estimate over the years 2015–2018. The color gradient shows the probability of establishment from 0 (white) to 1 (red). The bottom map represents the uncertainty of the prediction (higher values representing more uncertain predictions). Black dots show the positions where *Ae. albopictus* was established in 2019, which are used to compute the feature car distance to establishment (distance from the closest cell with establishment in the previous year). Maps modified from https://map.geoadmin.ch/ (accessed on 24 February 2022).

**Figure 5 ijerph-19-03220-f005:**
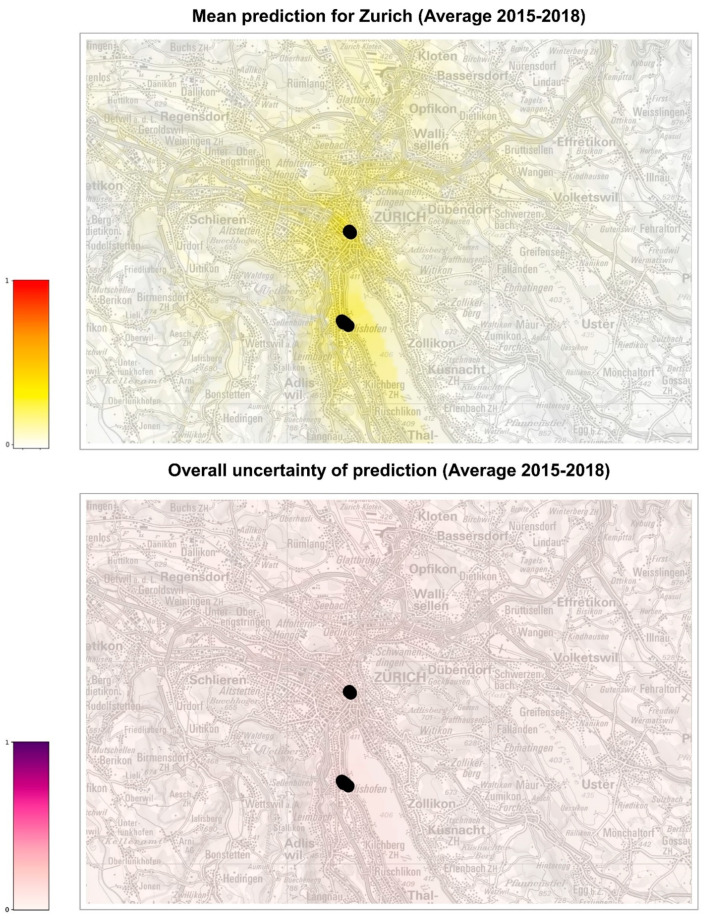
Probabilities of establishment of *Ae. albopictus* in Zurich. The map on top shows the average risk estimate over the years 2015–2018. The color gradient shows the probability of establishment from 0 (white) to 1 (red). The bottom map represents the uncertainty of the prediction (higher values representing more uncertain predictions). Black dots show the positions where *Ae. albopictus* was established in 2019, which are used to compute the feature car distance to establishment (distance from the closest cell with establishment in the previous year). Maps modified from https://map.geoadmin.ch/ (accessed on 24 February 2022).

**Table 1 ijerph-19-03220-t001:** Number of ovitraps analyzed in Ticino for the presence of *Ae. albopictus*, ovitraps with establishment of the mosquito, cells observed, and cells with establishment.

	2004	2005	2006	2007	2008	2009	2010	2011	2012	2013	2014	2015
Ovitraps analyzed	57	189	235	292	466	1241	1342	1357	1361	1389	1022	1031
Ovitraps established	3	0	5	23	94	144	181	265	580	524	463	793
Cells observed	25	76	82	101	163	887	973	983	1017	1108	925	928
Cells established	3	0	3	16	44	106	144	211	466	456	425	727

**Table 2 ijerph-19-03220-t002:** Informative predictors selected by lasso regularization.

Predictor	Description (Unit of Measure)	Frequency ^1^	Sign ^2^
elevation	Altitude (m.a.s.l.)	1.000	−
warm-s Tmin p5	5th percentile of the minimum temperature in the warm season (°C)	1.000	−
warm-s RAIN average	Average of precipitations in the warm season (mm)	1.000	−
cold-s Tmax p25	25th percentile of the maximum temperature in the cold season (°C)	1.000	+
car distance to establishment	Road-based distance in minutes of travel by car from the nearest cell established in the previous year (min)	1.000	−
clc121 (industrial) land cover	Percentage of industrial or commercial units covering a cell	1.000	−
Tmin minimum biweekly average	Average daily minimum temperature (Tmin) observed during the two-week period of the year with the lowest average Tmin (°C)	0.875	−
warm-s RAIN minimum biweekly average	Average precipitation observed during the two-week period of the warm season with the lowest average precipitation (mm)	0.875	+
cold-s RAIN p75	75th percentile of the precipitations in the cold season (mm)	0.750	+
warm-s Tmax p75	75th percentile of the maximum temperature in the warm season (°C)	0.500	+

^1^ Frequency: fraction of individual models of the ensemble including the feature. ^2^ Sign: sign of the feature coefficient.

**Table 3 ijerph-19-03220-t003:** Prediction performance of the regularized logistic regression in cross validation and on the test dataset. The area under the receiver operating curve (AUC) represents the probability that, given two random cells, one established and one non-established, the classifier assigns a larger risk of establishment to the established cell.

	Cross Validation ^1^ (2007–2012)	2013	2014	2015
Mean and standard deviation of the AUCs of the individual models of the ensemble ^2^	0.856 (0.047)	0.670 (0.063)	0.731 (0.008)	0.740 (0.007)
AUC of the ensemble average prediction		0.748	0.733	0.741

^1^ Cross validation considers only the six models for which the left-out-year used for validation has enough established cells (namely, years 2007 to 2012). ^2^ Standard deviations (SDs) are presented in parentheses.

## Data Availability

Data was obtained from the different sources cited in the text and is available with the permission of these sources.

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
