# Peer review of "Risk-Based Mapping Tools for Surveillance and Control of the Invasive Mosquito Aedes albopictus in Switzerland"

_ijerph, 2022, doi:10.3390/ijerph19063220_

Round 1
Reviewer 1 Report
The manuscript addresses the modelling work on identification of the most suitable areas in Switzerland for spread and establishment of Aedes albopictus, an important vector for viruses. The study is of great interest and the results presented are significant and informative, including the possible spatial spread of the species in several important cities in the country. The paper is well written, and the goals are well defined. The input data used for the model (entomological data and predictors) is appropriate and accurate, the methods applied are sound and the results are robust.
There are some minor issues I’d like to point out in hope they help the authors to improve the quality of the work:
Introduction:
Lines 64-65: I do not completely agree with the sentence ‘Ovitrap data indicate suitability of a habitat for oviposition and therefore for the establishment of populations”. Eggs deployed during summer in for example Scandinavian regions probably will not succeed to establish populations. Please give some nuance as for example “Ovitrap data indicate suitability of a habitat for oviposition and therefore for the possibility of establishment of populations”.
Line 77: percentage of total human population sounds odd used here. It will be better to present the % of the area of Ticino covered by surveillance and control instead of human population data.
Methods
Line 145: Please add a reference after 20C.
Line 151: Please add information of the software or method used for the rasterization.
Line 167: “ and Ae. albopictus eggs are counted.” I suppose you cannot distinguish Ae. albopictus eggs from other Aedes species only using a stereo microscope. How do you overcome this problem? Could you provide the keys used?
Lines 168-171: Please rephrase the sentence making the information clear for the reader.
Lines 175-176: Could the new species not be already present before 2015, and not detected? Please elaborate in the discussion section.
Lines 184-187: Same as before. Please rephrase or split the sentence to make it very clear for the reader. It is very difficult to follow.
Line 192-193: Move to results as this information results from the application of your method.
Results
Figure 1: Cell established (in grey) are not in the map. Please remove this or add the grey dots.
Figures 3, 4 and 5: I suggest to use only the average 2015-2018 as figure. Other individual maps could be available as supplementary material.
Author Response
The manuscript addresses the modelling work on identification of the most suitable areas in Switzerland for spread and establishment of Aedes albopictus, an important vector for viruses. The study is of great interest and the results presented are significant and informative, including the possible spatial spread of the species in several important cities in the country. The paper is well written, and the goals are well defined. The input data used for the model (entomological data and predictors) is appropriate and accurate, the methods applied are sound and the results are robust.
There are some minor issues I’d like to point out in hope they help the authors to improve the quality of the work:
Introduction:
Lines 64-65: I do not completely agree with the sentence ‘Ovitrap data indicate suitability of a habitat for oviposition and therefore for the establishment of populations”. Eggs deployed during summer in for example Scandinavian regions probably will not succeed to establish populations. Please give some nuance as for example “Ovitrap data indicate suitability of a habitat for oviposition and therefore for the possibility of establishment of populations”.
Reply: Sentence modified according to the reviewer’s suggestion (now lines 71-72).
Line 77: percentage of total human population sounds odd used here. It will be better to present the % of the area of Ticino covered by surveillance and control instead of human population data.
Reply: The canton of Ticino is a mountainous region where 50% of the territory in covered by woods and 30% by other natural surfaces. Only 5% of the territory is covered by settlements, where the tiger mosquito lives and the surveillance and control take place. Therefore, talking about percentage of the area of Ticino covered by surveillance and control does not seem the best option for this specific region. We would prefer to keep the original sentence (now lines 83-85).
Methods
Line 145: Please add a reference after 2°C.
Reply: Reference added (now line 155).
Line 151: Please add information of the software or method used for the rasterization.
Reply: We added the information of the softwares used (now line 167).
Line 167: “and Ae. albopictus eggs are counted.” I suppose you cannot distinguish Ae. albopictus eggs from other Aedes species only using a stereo microscope. How do you overcome this problem? Could you provide the keys used?
Reply: Yes, the reviewer is right. Other methods, such as rearing of adults and molecular analysis, were used to regularly control the morphological identification of eggs. They are described in our previous publications on the monitoring system. We modified the sentence to clarify this and to reference the previous work (now lines 176-186).
Lines 168-171: Please rephrase the sentence making the information clear for the reader.
Reply: The sentence was removed (now lines 187-190).
Lines 175-176: Could the new species not be already present before 2015, and not detected? Please elaborate in the discussion section.
Reply: Please see the previous reply to comment for line 167. From the different methods used to identify invasive Aedes species in Ticino and in Switzerland, we know that only after 2015 the two other invasive species Ae. koreicus and Ae. japonicus expanded enough in Ticino to potentially bring a bias in the monitoring of tiger mosquito. Data after 2015 were used for this reason. This is described in lines 176-186.
Lines 184-187: Same as before. Please rephrase or split the sentence to make it very clear for the reader. It is very difficult to follow.
Reply: The sentence has been rephrased and modified according to the suggestions of both reviewers (now lines 201-207).
Line 192-193: Move to results as this information results from the application of your method.
Reply: Lines 192-193 (now lines 212-213) are connected to Table 1 and the following paragraph (now lines 218-224). We didn’t understand if the reviewer would like us to move to the results only lines 192-193 (now 212-213) or the entire paragraph following these lines, including Table 1. Lines 198-204 (now 218-224) refer to Table 1 and tell the reader how many data were used in the model. If possible, we would like to keep also lines 192-193 (now 212-213) in the methods.
Results
Figure 1: Cell established (in grey) are not in the map. Please remove this or add the grey dots.
Reply: Figure 1 has been modified according to the suggestions of both reviewers. The legend with the grey dots has been removed.
Figures 3, 4 and 5: I suggest to use only the average 2015-2018 as figure. Other individual maps could be available as supplementary material.
Reply: We keep only the average 2015-2018 in the main text and move the individual maps to the supplementary material. Please see also the modifications made according to the suggestions of the other reviewer.
Reviewer 2 Report
Ravasi et al. have used almost a decade's worth of surveillance data to model the presence of Ae. albopictus in Switzerland, identifying regions at high risk for future spread.
I enjoyed reading this paper, and I found it interesting and timely. I thought the discussion in particular was excellent. I have a number of suggestions.
MAJOR CONCERNS:
1) Line 284: I assume that your predictor variables were scaled somehow before you started predictor selection and model fitting, etc. I can’t find anywhere where you say this, however. If you didn’t scale your data, you need to justify this (and I am finding it pretty difficult to image what possible justification you could have for that).
2) Lines 284-303: I am also struggling to figure out how you actually chose a model for each single “fold”. Perhaps I misunderstand LASSO, but as far as I can tell it is just a method for getting a correlation-coefficient-like-thing that penalizes having lots of predictors. I suppose the ideal thing to do would be get a LASSO number for every possible model and pick the best, but presumably you didn’t run 2^79 models. I suspect that what you did was some sort of add-one-variable-at-a-time method where you stopped when the LASSO results for the new model were no longer better than the previous one. But you never explicitly say this, and you don’t give us any idea of how you decided which order to add the variables, etc. I don’t suspect that you did anything statistically awful here, but I just have no way of evaluating this with the current information.
It’s possible that if my ML knowledge was a bit less patchy, so that I was familiar with LASSO and had more scikit-learn experience generally, I would know exactly what you did here. But I don’t think that gets you off the hook. I think that I am representative of a class of reader who will know enough about modelling to be very interested indeed in how you got from 2^79 parameter sets to 1 parameter set, but doesn’t know enough about ML to extract the details from what you have written.
MODERATE CONCERNS:
Lines 70-77: Has this monitoring scheme already been described in a publication? I think so, right (13 and 33?)? Please cite those sources here, if so.
Lines 115-120: To what extent does your approach actually “provide the required flexibility to handle complex and multiple interacting elements”? It seems like you really only use ML stuff to prune your set of predictor variables, and after that you are just doing regular old linear modelling. This isn’t bad—I’m a fan of regular old linear modelling—but I think you are setting up your reader to misunderstand what you actually did. For instance, this sentence made me think that you were going to do some type of random-forest-y kind of approach that would let you put local interactions among variables into your predictive model. I might just tone down the language here a smidge.
Lines 139-143: Does Ae. albopictus diapause in either or both of these regions of Switzerland? If so, when? I think that this will be of interest to anybody who works on Ae. albopictus specifically, but I also suspect it will help your definitions of “cold” and “warm” seasons seem a lot less arbitrary.
Line 200, and everywhere else: I do not want to see any of your python variable names (like “carD”, or the even worse “minMavgTmin”, etc. Get rid of all of these. They are very disorienting to the reader and can be replaced by informative names in every case. I make more detailed suggestions about what to do here in my comments for Table 2.
Lines 364-373: I think you need to give us more information on your predictor variables. To be clear, the additional information should go to the SI, not the main text. But people who work on Ae. albopictus especially will want to see:
First, a table giving the correlation between each pair of the variables shown in Table 2. I would like to know how good a job your ML method did of selecting variables that are not correlated especially strongly with each other.
Second, I would like to see the non-truncated version of Table 2. Are there 2 variables that showed up in <4 models? 20 variables? 69 variables? How I think about your set of predictors varies depending on how this question is answered.
Third, I would like to know, for each of your 10 chosen predictors, which of the other 78 variables were correlated strongly with it, for some arbitrarily-chosen definition of “strong”, like r=0.9 or something like that. You mention in the paper, quite rightly, that the chosen predictors probably represent suites of closely correlated variables. It would be good to know what those variables are!
Figures 3-5: I don’t like these. I have given some suggestions in the figures and tables section below.
Lines 676-677: I would strongly recommend that you make a dryad repository (or similar) with the raw data that you used from each source as well as all your analysis scripts. I think that this is best practice for all studies, but it is especially the case for this one, where you rely on a lot of other people’s data, but you often just cite a website that could potentially change whenever. I have real worries about whether this analysis is going to be repeatable 5 years from now if you don’t include your data.
MINOR CONCERNS:
Line 60: I recommend referencing Figure 1 here, as it is helpful for the reader to get a visualization.
Line 70: “Goods handling stations”—I am not sure what this means. It will not be idiomatic for your American readers, at least.
Line 119: I wasn’t sure what “with different outcomes on the most informative predictors” means.
Lines 143-150: Do we need to be given this information? If you are looking to cut stuff (a question for the editor, but this paper is LONG), this would be a good place to start.
Lines 173-174: I recommend just making a citation for this. The fact that this institution is called “info fauna – Swiss Centre for Cartography of the Fauna” is not your fault, but it is a baffling name that makes this sentence very hard to parse, and I assumed that it was a typo until I went to their webpage. Also, if the data is available online, you should link to that directly, not to the home page.
Lines 183-187: What is your logic for using 28 days as a cutoff? Might be worth including for the reader here.
Lines 212-216: The names for the different categories and the order in which they are presented is not consistent between his passage and Table S1.
Line 218: You should give a format citation here (or if the citations later in the paragraph suffice, just cut “(source: MeteoSwiss)”.
Line 251: I had to look up “aspect (i.e., land orientation)”. I would describe it as “aspect (i.e., the compass direction faced by sloping terrain” or similar.
Lines 262-264: You should give a formal citation here.
Lines 290-291: I do not know what “returns the probability relative to the class Established” means.
Lines 294-296: Doesn’t AIC do the thing you want? Why use LASSO over it? Is LASSO better analytically? Is it more tractable computationally? If it’s the former, you might want to mention that, since it would be a selling point for your method.
Lines 300-303: I think you should say something like “In a preliminary analysis, we found that inside the range [0.01,1] …”, as that will make it clear that you are not presenting those results here.
Line 322: I thought that your explanation in the discussion surrounding this decision was excellent (479-494). It might be worth referring to that part of the discussion here, so that any readers who are a bit unsatisfied by this explanation (I was until I got to your explanation in the discussion) can see that you have thought about this in some detail.
Lines 590-618: I wonder whether there is a way to show this visually? Perhaps a map like in Figure 3 that shows your predictions for 2015-2018, and then has circles around all the areas that actually got a new establishment in 2015-2018?
Lines 617-618: Presumably you’ve already alerted the authorities, so you can probably cut that clause and just leave the recommendation for surveillance.
Line 701: You said two week windows above: should this say “14” instead of “15”?
Appendix B: You never referred to this in the text. I was confused about what this meant and why you were showing it. If you aren’t going to refer to it in the main body, I suggest deleting it. Also, if you do keep it, you need to change the names of the predictors, and you should tell us what the sample sizes are for each box and whisker.
GRAMMER STUFF:
Feel free to ignore any/all of this, but your prose will be better for including it.
Line 78: “… Ae. albopictus has also been monitored since 2013…” would be better.
Line 102: I think it should be “socioenvironmental” or “socio-environmental”.
Line 111: Should be “... distribution of Ae. albopictus …”
Line 115: I would say “… modelling approaches like machine learning …”
Line 125: Should be “… whole of Switzerland …”
Line 127: I would say “… a small number of occurrences are available…”. Both are grammatically correct, but mine sounds better.
Line 128: I would say “… establishment intended to help…”
Line 255: I wouldn’t say “there are”—somebody could define them differently. Maybe move this sentence later in the paragraph and say “CORINE has divided Switzerland into 32 …” or similar.
Line 288: I think you should replace “learned” or “learned on” with “trained” as a description of models developed from training data; this occurs many times throughout the paper.
Line 307: “between” should be “among” instead.
Line 538: I would write “Two other selected predictors …”
FIGURE AND TABLE COMMENTS:
Figure 1:
The box on the map of Switzerland that shows the region included on the Ticino map is not correct; your map extends further east than the box would suggest.
The cells-established/cells-not-established thing isn’t working. From this figure, it looks like 90+% of cells are established, but we know this isn’t the case from Table 1; much less than half are. If it isn’t important for us to be able to see whether cells are established or not, then just fill in all the points and don’t show us which cells are established and which are not (this would be what I lean toward). But if it is important, this isn’t working and you need to make some change. I’m not sure what that change is; it probably involves splitting the figure into multiple facets.
Table 2:
As I mention above, these predictor names are gawdawful. I like the descriptions, but I can see why you might think that they are too unwieldy to use as predictor names in the text. I think there is a comprise, though, where you can also come up with 1-3 word predictor names to use instead of these very cryptic alphanumerics. For instance, you could use “Elevation”, “Spring Temperature”, “Summer Precipitation”, “Winter Temperature”, “Distance to Established Cell”, “Winter Severity”, “Summer Drought”, “Winter Precipitation”, “Summer Temperature”. Those are all off the top of my head in 30 seconds based only on what I know about these variables from your discussion, so probably you will come up with better names. As long as you make sure to mention (probably in the figure legend) that the predictor names represent your hypotheses about what each specific metric represents, I think you’re good.
Also, I think you should explain in the caption why sign has a number involved, rather than just a + or -. Presumably it is to include cases were models disagreed on the sign, but I am not 100% certain.
Figure 2:
Make sure to change the predictor names here as well.
Why are some of the coefficients zero? minMavgTmin, cRAIN_p75, and wTmax_p75 all have coefficients that seem EXTREMELY close to zero, which I struggle to explain given your model selection protocol.
Table 3:
I recommend having a description in the caption of what AUC is. This is in the text somewhere, but it is worth having it here for people who are just skimming the figures and tables (which will be a lot of your readers).
Figure 3:
First off, you should have the left and right columns have a different color scale. That will make it much clearer that you are looking at two different things.
Second, I cannot see the points. They need to be about 5X larger, AT LEAST. Since their color conveys information, they have to be large enough for us to observe the color easily.
Third, I think you need to go into a bit more detail about what “normalized standard deviation” is. Is this a linear transform? Or a z-score that then got transformed a second time? The scale makes me thing the former, but “normalized” threw me for a loop. Also, I’m pretty sure that you used normalized once across each year, and then used the same values for Figure 3, 4, and 5, but I’m not 100% sure that this is true, and you need to say that in the figure captions if it is (or if it isn’t).
Finally, are we SURE all these facets are necessary? I recommend leaving the ones for individual years for the SI. I would just show the two facets for the average, and then you can make them a lot bigger so that we can actually examine the details.
If you got rid of 8 of these facets, you might also consider getting rid of the tiny unreadable dots altogether and having a 3rd facet that is just the canton map with the (much, much larger) dots overlaid on it.
Figure 4:
All the same comments from Figure 3.
Also, having a road/political map on the background is both ugly as sin and making it hard for me to examine your heatmap. I recommend removing the political map from the background of each facet, and giving it its own separate facet. Plus, then we could actually read the labels of the political map. Right now you have the worst of both worlds.
Figure 5:
All the same comments from Figure 4.
Author Response
Ravasi et al. have used almost a decade's worth of surveillance data to model the presence of Ae. albopictus in Switzerland, identifying regions at high risk for future spread.
I enjoyed reading this paper, and I found it interesting and timely. I thought the discussion in particular was excellent. I have a number of suggestions.
MAJOR CONCERNS:
1) Line 284: I assume that your predictor variables were scaled somehow before you started predictor selection and model fitting, etc. I can’t find anywhere where you say this, however. If you didn’t scale your data, you need to justify this (and I am finding it pretty difficult to image what possible justification you could have for that).
Reply: The predictor variables were scaled using min-max normalization. We have added this information to section 2.4 of the paper (now lines 335-336).
2) Lines 284-303: I am also struggling to figure out how you actually chose a model for each single “fold”. Perhaps I misunderstand LASSO, but as far as I can tell it is just a method for getting a correlation-coefficient-like-thing that penalizes having lots of predictors. I suppose the ideal thing to do would be get a LASSO number for every possible model and pick the best, but presumably you didn’t run 2^79 models. I suspect that what you did was some sort of add-one-variable-at-a-time method where you stopped when the LASSO results for the new model were no longer better than the previous one. But you never explicitly say this, and you don’t give us any idea of how you decided which order to add the variables, etc. I don’t suspect that you did anything statistically awful here, but I just have no way of evaluating this with the current information.
It’s possible that if my ML knowledge was a bit less patchy, so that I was familiar with LASSO and had more scikit-learn experience generally, I would know exactly what you did here. But I don’t think that gets you off the hook. I think that I am representative of a class of reader who will know enough about modelling to be very interested indeed in how you got from 2^79 parameter sets to 1 parameter set, but doesn’t know enough about ML to extract the details from what you have written.
Reply: In our work we have not used LASSO to produce a goodness of fit measure of our models but to learn the models themselves. LASSO regression performs variable selection while learning the model, thus removing the need to test all possible subsets of the 79 features available. This is possible because the penalty component in the LASSO objective function forces the regression coefficient of the less relevant variables to zero. Therefore, for each training dataset, we have learned one single (regularized) logistic regression model including all the 79 predictors, but, having used a large enough regularization coefficient, most of the predictors were ignored (i.e., assigned a null coefficient) in the learned model.
This has been better explained in section 2.4 of the paper (now lines 340-346 and lines 383-390).
MODERATE CONCERNS:
Lines 70-77: Has this monitoring scheme already been described in a publication? I think so, right (13 and 33?)? Please cite those sources here, if so.
Reply: Reference added (now lines 80-83).
Lines 115-120: To what extent does your approach actually “provide the required flexibility to handle complex and multiple interacting elements”? It seems like you really only use ML stuff to prune your set of predictor variables, and after that you are just doing regular old linear modelling. This isn’t bad—I’m a fan of regular old linear modelling—but I think you are setting up your reader to misunderstand what you actually did. For instance, this sentence made me think that you were going to do some type of random-forest-y kind of approach that would let you put local interactions among variables into your predictive model. I might just tone down the language here a smidge.
Reply: We modified the sentence to tone down the language ( now lines 123-125).
Lines 139-143: Does Ae. albopictus diapause in either or both of these regions of Switzerland? If so, when? I think that this will be of interest to anybody who works on Ae. albopictus specifically, but I also suspect it will help your definitions of “cold” and “warm” seasons seem a lot less arbitrary.
Reply: In Switzerland, Ae. albopictus diapauses in the winter months. The females start to lay diapausing eggs in September and eggs remain in diapause until mid- to end- April. We added this information to the paragraph on the description of cold and warm seasons in section 2.3 (now lines 264-266 and 267-268).
Line 200, and everywhere else: I do not want to see any of your python variable names (like “carD”, or the even worse “minMavgTmin”, etc. Get rid of all of these. They are very disorienting to the reader and can be replaced by informative names in every case. I make more detailed suggestions about what to do here in my comments for Table 2.
Reply: The abbreviation have been replaced by more informative ones. Please see our reply in the reviewer’s comments for Table 2.
Lines 364-373: I think you need to give us more information on your predictor variables. To be clear, the additional information should go to the SI, not the main text. But people who work on Ae. albopictus especially will want to see:
- First, a table giving the correlation between each pair of the variables shown in Table 2. I would like to know how good a job your ML method did of selecting variables that are not correlated especially strongly with each other.
Reply: The information has been added to Appendix B.
- Second, I would like to see the non-truncated version of Table 2. Are there 2 variables that showed up in <4 models? 20 variables? 69 variables? How I think about your set of predictors varies depending on how this question is answered.
Reply: As mentioned at lines 349-351, the LASSO regularization coefficient is chosen so that the models include a maximum of 14 variables. We have added the information “overall, only 17 variables are selected by at least one model.” To section 2.4 of the manuscript (now line 351).
We believe it would be misleading to also show variables that are selected by few models, as those are not likely to be actually related to mosquitoes’ establishment.
- Third, I would like to know, for each of your 10 chosen predictors, which of the other 78 variables were correlated strongly with it, for some arbitrarily-chosen definition of “strong”, like r=0.9 or something like that. You mention in the paper, quite rightly, that the chosen predictors probably represent suites of closely correlated variables. It would be good to know what those variables are!
Reply: The information about the degree of correlation between predictors has been added to Appendix B.
Figures 3-5: I don’t like these. I have given some suggestions in the figures and tables section below.
Reply: Please see our reply in the specific comment on Figures 3-5.
Lines 676-677: I would strongly recommend that you make a dryad repository (or similar) with the raw data that you used from each source as well as all your analysis scripts. I think that this is best practice for all studies, but it is especially the case for this one, where you rely on a lot of other people’s data, but you often just cite a website that could potentially change whenever. I have real worries about whether this analysis is going to be repeatable 5 years from now if you don’t include your data.
Reply: Except for the data on tiger mosquito in Ticino, all the raw data used for this study falls under the category of third-party data. Therefore, restrictions apply to the availability of this data. This is made clear in the Data Availability Statement and the sources are cited within the main text. Data on tiger mosquito in Ticino and Switzerland are sourced from info fauna, which is a Swiss federal office. We don’t think that there is a risk of this office to change whenever. The data on mosquito is available from this office under request, as written in the text.
MINOR CONCERNS:
Line 60: I recommend referencing Figure 1 here, as it is helpful for the reader to get a visualization.
Reply: Reference to Figure 1 inserted.
Line 70: “Goods handling stations”—I am not sure what this means. It will not be idiomatic for your American readers, at least.
Reply: we substituted the term with “freight sorting facilities” (now line 77).
Line 119: I wasn’t sure what “with different outcomes on the most informative predictors” means.
Reply: we modified the sentence to clarify the meaning (now lines 125-129).
Lines 143-150: Do we need to be given this information? If you are looking to cut stuff (a question for the editor, but this paper is LONG), this would be a good place to start.
Reply: We would like to keep this information on how climate change is affecting the study area. In our opinion it adds important information for the discussion on the predictions of establishment of the tiger mosquito.
Lines 173-174: I recommend just making a citation for this. The fact that this institution is called “info fauna – Swiss Centre for Cartography of the Fauna” is not your fault, but it is a baffling name that makes this sentence very hard to parse, and I assumed that it was a typo until I went to their webpage. Also, if the data is available online, you should link to that directly, not to the home page.
Reply: We shortened the name to “info fauna – CSCF". The data is available under request, so we cannot link directly to a specific page but only to the home page (now lines 192-193).
Lines 183-187: What is your logic for using 28 days as a cutoff? Might be worth including for the reader here.
Reply: The 28 days cutoff was defined in order to take into account the fact that from 2013 slats were analysed at alternate rounds (i.e., every 28 days). We modified the sentence to better explain this (now lines 201-207).
Lines 212-216: The names for the different categories and the order in which they are presented is not consistent between this passage and Table S1.
Reply: We modified the order of the categories in Table S1 according to the order in which they are presented in this sentence. We also used the same order for the description of the categories in the methods.
Line 218: You should give a formal citation here (or if the citations later in the paragraph suffice, just cut “(source: MeteoSwiss)”.
Reply: Here we are following the indications on the conditions of use specified by MeteoSwiss and therefore we think that we have to keep it as we have written it.
Line 251: I had to look up “aspect (i.e., land orientation)”. I would describe it as “aspect (i.e., the compass direction faced by sloping terrain” or similar.
Reply: we modified the sentence according to the reviewer’s suggestion (now lines 238-239).
Lines 262-264: You should give a formal citation here.
Reply: Here too, we are following the indications on the conditions of use specified by the Swiss Federal Statistical Office and therefore we think that we should keep it as it is written.
Lines 290-291: I do not know what “returns the probability relative to the class Established” means.
Reply: Rephrased as “returns the probability that variable Established takes value TRUE” (now lines 333-335).
Lines 294-296: Doesn’t AIC do the thing you want? Why use LASSO over it? Is LASSO better analytically? Is it more tractable computationally? If it’s the former, you might want to mention that, since it would be a selling point for your method.
Reply: AIC is an estimator of the prediction error that can be used to compare different models trained using different predictors and select the best one. LASSO is a method to automatically select the predictors while learning the model. Both approaches can be used to produce regression models with an optimal subset of features. We found LASSO an effective approach in our setting, but other methods could also be used. We expect them to produce similar results.
Lines 300-303: I think you should say something like “In a preliminary analysis, we found that inside the range [0.01,1] …”, as that will make it clear that you are not presenting those results here.
Reply: Thanks for the suggestion. We have implemented it in the paper (now lines 347-351).
Line 322: I thought that your explanation in the discussion surrounding this decision was excellent (479-494). It might be worth referring to that part of the discussion here, so that any readers who are a bit unsatisfied by this explanation (I was until I got to your explanation in the discussion) can see that you have thought about this in some detail.
Reply: To avoid the unsatisfaction of the reader at this point of the paper, we have tried to better motivate our validation strategy in section 2.4. (now lines 366-378).
Lines 590-618: I wonder whether there is a way to show this visually? Perhaps a map like in Figure 3 that shows your predictions for 2015-2018, and then has circles around all the areas that actually got a new establishment in 2015-2018?
Reply: We added a supplementary figure (Figure S4) with a map of Switzerland showing the cantons and the areas we talk about, so that the reader can refer to this map while reading the text.
Lines 617-618: Presumably you’ve already alerted the authorities, so you can probably cut that clause and just leave the recommendation for surveillance.
Reply: We modified the sentence according to the reviewer’s suggestion (now lines 715-717).
Line 701: You said two week windows above: should this say “14” instead of “15”?
Reply: We wrote “14” instead of “15” (now line 815).
Appendix B: You never referred to this in the text. I was confused about what this meant and why you were showing it. If you aren’t going to refer to it in the main body, I suggest deleting it. Also, if you do keep it, you need to change the names of the predictors, and you should tell us what the sample sizes are for each box and whisker.
Reply: We prefer to delete this information from Appendix B. Instead, we added more information on the correlation between predictors (please refer to the rewiever’s comment to lines 364-373). We referred to this Appendix B in the text.
GRAMMER STUFF:
Feel free to ignore any/all of this, but your prose will be better for including it.
Line 78: “… Ae. albopictus has also been monitored since 2013…” would be better.
Reply: Sentence modified accordingly (now line 86).
Line 102: I think it should be “socioenvironmental” or “socio-environmental”.
Reply: Sentence modified accordingly (now line 110).
Line 111: Should be “... distribution of Ae. albopictus …”
Reply: Sentence modified accordingly (now line 119).
Line 115: I would say “… modelling approaches like machine learning …”
Reply: Sentence modified accordingly (now line 123).
Line 125: Should be “… whole of Switzerland …”
Reply: Sentence modified accordingly (now line 134).
Line 127: I would say “… a small number of occurrences are available…”. Both are grammatically correct, but mine sounds better.
Reply: Sentence modified accordingly (now line 136).
Line 128: I would say “… establishment intended to help…”
Reply: Sentence modified accordingly (now line 137).
Line 255: I wouldn’t say “there are”—somebody could define them differently. Maybe move this sentence later in the paragraph and say “CORINE has divided Switzerland into 32 …” or similar.
Reply: Sentence modified accordingly (now lines 246-248).
Line 288: I think you should replace “learned” or “learned on” with “trained” as a description of models developed from training data; this occurs many times throughout the paper.
Reply: In fact, one should use “trained on” or “learned from” and not “learned on”. We have preferred “learned from” to avoid annoying repetition of the word “training”.
Line 307: “between” should be “among” instead.
Reply: Sentence modified accordingly (now line 355).
Line 538: I would write “Two other selected predictors …”
Reply: Sentence modified accordingly (now line 634).
FIGURE AND TABLE COMMENTS:
Figure 1:
The box on the map of Switzerland that shows the region included on the Ticino map is not correct; your map extends further east than the box would suggest.
The cells-established/cells-not-established thing isn’t working. From this figure, it looks like 90+% of cells are established, but we know this isn’t the case from Table 1; much less than half are. If it isn’t important for us to be able to see whether cells are established or not, then just fill in all the points and don’t show us which cells are established and which are not (this would be what I lean toward). But if it is important, this isn’t working and you need to make some change. I’m not sure what that change is; it probably involves splitting the figure into multiple facets.
Reply: Figure 1 has been modified according to the suggestions of both reviewers. The box on the map of Switzerland has been adjusted. The map has been split into two facets to show both observed and established cells more clearly.
Table 2:
As I mention above, these predictor names are gawdawful. I like the descriptions, but I can see why you might think that they are too unwieldy to use as predictor names in the text. I think there is a comprise, though, where you can also come up with 1-3 word predictor names to use instead of these very cryptic alphanumerics. For instance, you could use “Elevation”, “Spring Temperature”, “Summer Precipitation”, “Winter Temperature”, “Distance to Established Cell”, “Winter Severity”, “Summer Drought”, “Winter Precipitation”, “Summer Temperature”. Those are all off the top of my head in 30 seconds based only on what I know about these variables from your discussion, so probably you will come up with better names. As long as you make sure to mention (probably in the figure legend) that the predictor names represent your hypotheses about what each specific metric represents, I think you’re good.
Reply: We modified the abbreviated names of the variables in order to have more informative names. We tried to accomplish the task without losing the real meaning of the variable. We hope that we managed to improve the clarity of the table and the main text.
Also, I think you should explain in the caption why sign has a number involved, rather than just a + or -. Presumably it is to include cases were models disagreed on the sign, but I am not 100% certain.
Reply: The number 1.0 after the sign + or - has no real meaning so we left only the signs in Table 2.
Figure 2:
Make sure to change the predictor names here as well.
Reply: Thank you. We changed the predictor names here as well.
Why are some of the coefficients zero? minMavgTmin, cRAIN_p75, and wTmax_p75 all have coefficients that seem EXTREMELY close to zero, which I struggle to explain given your model selection protocol.
Reply: Coefficients are zero for the models which did not select those predictors. LASSO models shrink coefficients toward zero. Personally, we do not see why one should not expect any small coefficient.
Table 3:
I recommend having a description in the caption of what AUC is. This is in the text somewhere, but it is worth having it here for people who are just skimming the figures and tables (which will be a lot of your readers).
Reply: We added the description of AUC in the caption of Table 3.
Figure 3:
First off, you should have the left and right columns have a different color scale. That will make it much clearer that you are looking at two different things.
Reply: Good idea. We used two different color scales.
Second, I cannot see the points. They need to be about 5X larger, AT LEAST. Since their color conveys information, they have to be large enough for us to observe the color easily.
Reply: We enlarged the dots.
Third, I think you need to go into a bit more detail about what “normalized standard deviation” is. Is this a linear transform? Or a z-score that then got transformed a second time? The scale makes me thing the former, but “normalized” threw me for a loop. Also, I’m pretty sure that you used normalized once across each year, and then used the same values for Figure 3, 4, and 5, but I’m not 100% sure that this is true, and you need to say that in the figure captions if it is (or if it isn’t).
Reply: The procedure for the normalization of the standard deviation has been better explained in section 3.2.
Finally, are we SURE all these facets are necessary? I recommend leaving the ones for individual years for the SI. I would just show the two facets for the average, and then you can make them a lot bigger so that we can actually examine the details.
Reply: Following the suggestion of both reviewers, we kept the two facets for the average in the main text and left the ones for individual years for the SI.
If you got rid of 8 of these facets, you might also consider getting rid of the tiny unreadable dots altogether and having a 3rd facet that is just the canton map with the (much, much larger) dots overlaid on it.
Reply: We think that it is more useful to keep the dots on the same map with the predictions. But every should now be more readable with the modification of the maps.
Figure 4:
All the same comments from Figure 3.
Reply: We made the same modifications as for Figure 3 and 5.
Also, having a road/political map on the background is both ugly as sin and making it hard for me to examine your heatmap. I recommend removing the political map from the background of each facet, and giving it its own separate facet. Plus, then we could actually read the labels of the political map. Right now you have the worst of both worlds.
Reply: We think that it is very useful to keep the road map on the background. But we changed the colours of the road map so that there is no confusion with the colours of the prediction.
Figure 5:
All the same comments from Figure 4.
Reply: We made the same modifications as for Figure 3 and 4.